# Two-Target Algorithms for Infinite-Armed Bandits with Bernoulli Rewards

**Thomas Bonald**[*]
Department of Networking and Computer Science
Telecom ParisTech
Paris, France
thomas.bonald@telecom-paristech.fr

**Alexandre Proutière**[*†]
Automatic Control Department
KTH
Stockholm, Sweden
alepro@kth.se

## Abstract

We consider an infinite-armed bandit problem with Bernoulli rewards. The mean rewards are independent, uniformly distributed over $[0, 1]$. Rewards 0 and 1 are referred to as a success and a failure, respectively. We propose a novel algorithm where the decision to exploit any arm is based on two successive targets, namely, the total number of successes until the first failure and until the first $m$ failures, respectively, where $m$ is a fixed parameter. This two-target algorithm achieves a long-term average regret in $\sqrt{2n}$ for a large parameter $m$ and a known time horizon $n$. This regret is optimal and strictly less than the regret achieved by the best known algorithms, which is in $2\sqrt{n}$. The results are extended to any mean-reward distribution whose support contains 1 and to unknown time horizons. Numerical experiments show the performance of the algorithm for finite time horizons.

## 1 Introduction

**Motivation.** While classical multi-armed bandit problems assume a finite number of arms [9], many practical situations involve a large, possibly infinite set of options for the player. This is the case for instance of on-line advertisement and content recommandation, where the objective is to propose the most appropriate categories of items to each user in a very large catalogue. In such situations, it is usually not possible to explore all options, a constraint that is best represented by a bandit problem with an infinite number of arms. Moreover, even when the set of options is limited, the time horizon may be too short in practice to enable the full exploration of these options. Unlike classical algorithms like UCB [10, 1], which rely on a initial phase where all arms are sampled once, algorithms for infinite-armed bandits have an intrinsic *stopping rule* in the number of arms to explore. We believe that this provides useful insights into the design of efficient algorithms for usual finite-armed bandits when the time horizon is relatively short.

**Overview of the results.** We consider a stochastic infinite-armed bandit with Bernoulli rewards, the mean reward of each arm having a uniform distribution over $[0, 1]$. This model is representative of a number of practical situations, such as content recommandation systems with like/dislike feedback and without any prior information on the user preferences. We propose a two-target algorithm based on some fixed parameter $m$ that achieves a long-term average regret in $\sqrt{2n}$ for large $m$ and a large known time horizon $n$. We prove that this regret is optimal. The anytime version of this algorithm achieves a long-term average regret in $2\sqrt{n}$ for unknown time horizon $n$, which we conjecture to be also optimal. The results are extended to any mean-reward distribution whose support contains 1. Specifically, if the probability that the mean reward exceeds $u$ is equivalent to $\alpha(1 - u)^\beta$ when

---

[*]The authors are members of the LINCS, Paris, France. See www.lincs.fr.
[†]Alexandre Proutière is also affiliated to INRIA, Paris-Rocquencourt, France.

$u \to 1^-$, the two-target algorithm achieves a long-term average regret in $C(\alpha, \beta) n^{\frac{\beta}{\beta+1}}$, with some explicit constant $C(\alpha, \beta)$ that depends on whether the time horizon is known or not. This regret is provably optimal when the time horizon is known. The precise statements and proofs of these more general results are given in the appendix.

**Related work.** The stochastic infinite-armed bandit problem has first been considered in a general setting by Mallows and Robbins [12] and then in the particular case of Bernoulli rewards by Herschkorn, Peköz and Ross [6]. The proposed algorithms are *first-order* optimal in the sense that they minimize the ratio $R_n/n$ for large $n$, where $R_n$ is the regret after $n$ time steps. In the considered setting of Bernoulli rewards with mean rewards uniformly distributed over $[0, 1]$, this means that the ratio $R_n/n$ tends to 0 almost surely. We are interested in *second-order* optimality, namely, in minimizing the equivalent of $R_n$ for large $n$. This issue is addressed by Berry et. al. [2], who propose various algorithms achieving a long-term average regret in $2\sqrt{n}$, conjecture that this regret is optimal and provide a lower bound in $\sqrt{2n}$. Our algorithm achieves a regret that is arbitrarily close to $\sqrt{2n}$, which invalidates the conjecture. We also provide a proof of the lower bound in $\sqrt{2n}$ since that given in [2, Theorem 3] relies on the incorrect argument that the number of explored arms and the mean rewards of these arms are independent random variables[1]; the extension to any mean-reward distribution [2, Theorem 11] is based on the same erroneous argument[2].

The algorithms proposed by Berry et. al. [2] and applied in [11, 4, 5, 7] to various mean-reward distributions are variants of the 1-failure strategy where each arm is played until the first failure, called a *run*. For instance, the non-recalling $\sqrt{n}$-run policy consists in exploiting the first arm giving a run larger than $\sqrt{n}$. For a uniform mean-reward distribution over $[0, 1]$, the average number of explored arms is $\sqrt{n}$ and the selected arm is exploited for the equivalent of $n$ time steps with an expected failure rate of $1/\sqrt{n}$, yielding the regret of $2\sqrt{n}$. We introduce a second target to improve the expected failure rate of the selected arm, at the expense of a slightly more expensive exploration phase. Specifically, we show that it is optimal to explore $\sqrt{n/2}$ arms on average, resulting in the expected failure rate $1/\sqrt{2n}$ of the exploited arm, for the equivalent of $n$ time steps, hence the regret of $\sqrt{2n}$. For unknown horizon times, anytime versions of the algorithms of Berry et. al. [2] are proposed by Teytaud, Gelly and Sebag in [13] and proved to achieve a regret in $O(\sqrt{n})$. We show that the anytime version of our algorithm achieves a regret arbitrarily close to $2\sqrt{n}$, which we conjecture to be optimal.

Our results extend to any mean-reward distribution whose support contains 1, the regret depending on the characteristics of this distribution around 1. This problem has been considered in the more general setting of bounded rewards by Wang, Audibert and Munos [15]. When the time horizon is known, their algorithms consist in exploring a pre-defined set of $K$ arms, which depends on the parameter $\beta$ mentioned above, using variants of the UCB policy [10, 1]. In the present case of Bernoulli rewards and mean-reward distributions whose support contains 1, the corresponding regret is in $n^{\frac{\beta}{\beta+1}}$, up to logarithmic terms coming from the exploration of the $K$ arms, as in usual finite-armed bandits algorithms [9]. The nature of our algorithm is very different in that it is based on a stopping rule in the exploration phase that depends on the observed rewards. This does not only remove the logarithmic terms in the regret but also achieves the optimal constant.

## 2 Model

We consider a stochastic multi-armed bandit with an infinite number of arms. For any $k = 1, 2, \ldots$, the rewards of arm $k$ are Bernoulli with unknown parameter $\theta_k$. We refer to rewards 0 and 1 as a failure and a success, respectively, and to a *run* as a consecutive sequence of successes followed by a failure. The mean rewards $\theta_1, \theta_2, \ldots$ are themselves random, uniformly distributed over $[0, 1]$.

At any time $t = 1, 2, \ldots$, we select some arm $I_t$ and receive the corresponding reward $X_t$, which is a Bernoulli random variable with parameter $\theta_{I_t}$. We take $I_1 = 1$ by convention. At any time $t = 2, 3, \ldots$, the arm selection only depends on previous arm selections and rewards; formally, the random variable $I_t$ is $\mathcal{F}_{t-1}$-mesurable, where $\mathcal{F}_t$ denotes the $\sigma$-field generated by the set $\{I_1, X_1, \ldots, I_t, X_t\}$. Let $K_t$ be the number of arms selected until time $t$. Without any loss of generality, we assume that $\{I_1, \ldots, I_t\} = \{1, 2, \ldots, K_t\}$ for all $t = 1, 2, \ldots$, i.e., new arms are selected sequentially. We also assume that $I_{t+1} = I_t$ whenever $X_t = 1$: if the selection of arm $I_t$ gives a success at time $t$, the same arm is selected at time $t + 1$.

The objective is to maximize the cumulative reward or, equivalently, to minimize the regret defined by $R_n = n - \sum_{t=1}^{n} X_t$. Specifically, we focus on the average regret $E(R_n)$, where expectation is taken over all random variables, including the sequence of mean rewards $\theta_1, \theta_2, \ldots$. The time horizon $n$ may be known or unknown.

# 3 Known time horizon

## 3.1 Two-target algorithm

The two-target algorithm consists in exploring new arms until two successive targets $\ell_1$ and $\ell_2$ are reached, in which case the current arm is exploited until the time horizon $n$. The first target aims at discarding "bad" arms while the second aims at selecting a "good" arm. Specifically, using the names of the variables indicated in the pseudo-code below, if the length $L$ of the first run of the current arm $I$ is less than $\ell_1$, this arm is discarded and a new arm is selected; otherwise, arm $I$ is pulled for $m - 1$ additional runs and exploited until time $n$ if the total length $L$ of the $m$ runs is at least $\ell_2$, where $m \geq 2$ is a fixed parameter of the algorithm. We prove in Proposition 1 below that, for large $m$, the target values[3] $\ell_1 = \lfloor \sqrt[3]{\frac{n}{2}} \rfloor$ and $\ell_2 = \lfloor m\sqrt{\frac{n}{2}} \rfloor$ provide a regret in $\sqrt{2n}$.

---

**Algorithm 1:** Two-target algorithm with known time horizon $n$.

---
**Parameters:** $m, n$
**Function:**
*Explore*
$I \leftarrow I + 1, L \leftarrow 0, M \leftarrow 0$
**Algorithm:**
$\ell_1 = \lfloor \sqrt[3]{\frac{n}{2}} \rfloor, \ell_2 = \lfloor m\sqrt{\frac{n}{2}} \rfloor$
$I \leftarrow 0$
*Explore*
Exploit $\leftarrow$ **false**
**forall the** $t = 1, 2, \ldots, n$ **do**
    Get reward $X$ from arm $I$
    **if not** Exploit **then**
        **if** $X = 1$ **then**
            $L \leftarrow L + 1$
        **else**
            $M \leftarrow M + 1$
            **if** $M = 1$ **then**
                **if** $L < \ell_1$ **then**
                    *Explore*
            **else if** $M = m$ **then**
                **if** $L < \ell_2$ **then**
                    *Explore*
                **else**
                    Exploit $\leftarrow$ **true**

## 3.2 Regret analysis

**Proposition 1** *The two-target algorithm with targets $\ell_1 = \lfloor \sqrt[3]{\frac{n}{2}} \rfloor$ and $\ell_2 = \lfloor m\sqrt{\frac{n}{2}} \rfloor$ satisfies:*

$$\forall n \geq \frac{m^2}{2}, \quad E(R_n) \leq m + \frac{\ell_2 + 1}{m} \left( \frac{\ell_2 - m + 2}{\ell_2 - \ell_1 - m + 2} \right)^m \left( 2 + \frac{1}{m} + 2\frac{m+1}{\ell_1 + 1} \right).$$

*In particular,*

$$\limsup_{n \to +\infty} \frac{E(R_n)}{\sqrt{n}} \leq \sqrt{2} + \frac{1}{m\sqrt{2}}.$$

*Proof.* Note that Let $U_1 = 1$ if arm 1 is used until time $n$ and $U_1 = 0$ otherwise. Denote by $M_1$ the total number of 0's received from arm 1. We have:

$$E(R_n) \leq P(U_1 = 0)(E(M_1|U_1 = 0) + E(R_n)) + P(U_1 = 1)(m + nE(1 - \theta_1|U_1 = 1)),$$

so that:

$$E(R_n) \leq \frac{E(M_1|U_1 = 0)}{P(U_1 = 1)} + m + nE(1 - \theta_1|U_1 = 1). \tag{1}$$

Let $N_t$ be the number of 0's received from arm 1 until time $t$ when this arm is played until time $t$. Note that $n \geq \frac{m^2}{2}$ implies $n \geq \ell_2$. Since $P(N_{\ell_1} = 0|\theta_1 = u) = u^{\ell_1}$, the probability that the first target is achieved by arm 1 is given by:

$$P(N_{\ell_1} = 0) = \int_0^1 u^{\ell_1} \mathrm{d}u = \frac{1}{\ell_1 + 1}.$$

Similarly,

$$P(N_{\ell_2 - \ell_1} < m|\theta_1 = u) = \sum_{j=0}^{m-1} \binom{\ell_2 - \ell_1}{j} u^{\ell_2 - \ell_1 - j}(1 - u)^j,$$

so that the probability that arm 1 is used until time $n$ is given by:

$$
\begin{aligned}
P(U_1 = 1) &= \int_0^1 P(N_{\ell_1} = 0|\theta_1 = u)P(N_{\ell_2 - \ell_1} < m|\theta_1 = u)\mathrm{d}u, \\
&= \sum_{j=0}^{m-1} \frac{(\ell_2 - \ell_1)!}{(\ell_2 - \ell_1 - j)!} \frac{(\ell_2 - j)!}{(\ell_2 + 1)!}.
\end{aligned}
$$

We deduce:

$$\frac{m}{\ell_2 + 1} \left( \frac{\ell_2 - \ell_1 - m + 2}{\ell_2 - m + 2} \right)^m \leq P(U_1 = 1) \leq \frac{m}{\ell_2 + 1}. \tag{2}$$

Moreover,

$$E(M_1|U_1 = 0) = 1 + (m - 1)P(N_{\ell_1} = 0|U_1 = 0) \leq 1 + (m - 1)\frac{P(N_{\ell_1} = 0)}{P(U_1 = 0)} \leq 1 + 2\frac{m+1}{\ell_1 + 1},$$

where the last inequality follows from (2) and the fact that $\ell_2 \geq \frac{m^2}{2}$.

It remains to calculate $E(1 - \theta_1|U_1 = 1)$. Since:

$$P(U_1 = 1|\theta_1 = u) = \sum_{j=0}^{m-1} \binom{\ell_2 - \ell_1}{j} u^{\ell_2 - j}(1 - u)^j,$$

we deduce:

$$
\begin{aligned}
E(1 - \theta_1|U_1 = 1) &= \frac{1}{P(U_1 = 1)} \int_0^1 \sum_{j=0}^{m-1} \binom{\ell_2 - \ell_1}{j} u^{\ell_2 - j}(1 - u)^{j+1}\mathrm{d}u, \\
&= \frac{1}{P(U_1 = 1)} \sum_{j=0}^{m-1} \frac{(\ell_2 - \ell_1)!}{(\ell_2 - \ell_1 - j)!} \frac{(\ell_2 - j)!}{(\ell_2 + 2)!}(j + 1), \\
&\leq \frac{1}{P(U_1 = 1)} \frac{m(m+1)}{2(\ell_2 + 1)(\ell_2 + 2)} \leq \frac{1}{P(U_1 = 1)} \left( 1 + \frac{1}{m} \right).
\end{aligned}
$$

The proof then follows from (1) and (2). $\quad\square$

### 3.3 Lower bound

The following result shows that the two-target algorithm is asymptotically optimal (for large $m$).

**Theorem 1** *For any algorithm with known time horizon $n$,*

$$\liminf_{n \to +\infty} \frac{E(R_n)}{\sqrt{n}} \geq \sqrt{2}.$$

*Proof.* We present the main ideas of the proof. The details are given in the appendix. Assume an oracle reveals the parameter of each arm after the first failure of this arm. With this information, the optimal policy explores a random number of arms, each until the first failure, then plays only one of these arms until time $n$. Let $\mu$ be the parameter of the best known arm at time $t$. Since the probability that any new arm is better than this arm is $1 - \mu$, the mean cost of exploration to find a better arm is $\frac{1}{1-\mu}$. The corresponding mean reward has a uniform distribution over $[\mu, 1]$ so that the mean gain of exploitation is less than $(n - t)\frac{1-\mu}{2}$ (it is not equal to this quantity due to the time spent in exploration). Thus if $1 - \mu < \sqrt{\frac{2}{n-t}}$, it is preferable not to explore new arms and to play the best known arm, with mean reward $\mu$, until time $n$. A fortiori, the best known arm is played until time $n$ whenever its parameter is larger than $1 - \sqrt{\frac{2}{n}}$. We denote by $A_n$ the first arm whose parameter is larger than $1 - \sqrt{\frac{2}{n}}$. We have $K_n \leq A_n$ (the optimal policy cannot explore more than $A_n$ arms) and

$$E(A_n) = \sqrt{\frac{n}{2}}.$$

The parameter $\theta_{A_n}$ of arm $A_n$ is uniformly distributed over $[1 - \sqrt{\frac{2}{n}}, 1]$, so that

$$E(\theta_{A_n}) = 1 - \sqrt{\frac{1}{2n}}. \tag{3}$$

For all $k = 1, 2, \ldots$, let $L_1(k)$ be the length of the first run of arm $k$. We have:

$$E(L_1(1) + \ldots + L_1(A_n - 1)) = (E(A_n) - 1)E(L_1(1)|\theta_1 \leq 1 - \sqrt{\frac{2}{n}}) = (\sqrt{\frac{n}{2}} - 1)\frac{-\ln(\sqrt{\frac{2}{n}})}{1 - \sqrt{\frac{2}{n}}}, \tag{4}$$

using the fact that:

$$E(L_1(1)|\theta_1 \leq 1 - \sqrt{\frac{2}{n}}) = \int_0^{1-\sqrt{\frac{2}{n}}} \frac{1}{1-u} \frac{du}{1 - \sqrt{\frac{2}{n}}}.$$

In particular,

$$\lim_{n \to +\infty} \frac{1}{n} E(L_1(1) + \ldots + L_1(A_n - 1)) \to 0 \tag{5}$$

and

$$\lim_{n \to +\infty} \frac{1}{n} P(L_1(1) + \ldots + L_1(A_n - 1) \leq n^{\frac{4}{5}}) \to 1.$$

To conclude, we write:

$$E(R_n) \geq E(K_n) + E((n - L_1(1) - \ldots - L_1(A_n - 1)))(1 - \theta_{A_n})).$$

Observe that, on the event $\{L_1(1) + \ldots + L_1(A_n - 1) \leq n^{\frac{4}{5}}\}$, the number of explored arms satisfies $K_n \geq A'_n$ where $A'_n$ denotes the first arm whose parameter is larger than $1 - \sqrt{\frac{2}{n-n^{\frac{4}{5}}}}$. Since $P(L_1(1) + \ldots + L_1(A_n - 1) \leq n^{\frac{4}{5}}) \to 1$ and $E(A'_n) = \sqrt{\frac{n-n^{\frac{4}{5}}}{2}}$, we deduce that:

$$\liminf_{n \to +\infty} \frac{E(K_n)}{\sqrt{n}} \geq \frac{1}{\sqrt{2}}.$$

By the independence of $\theta_{A_n}$ and $L_1(1), \ldots, L_1(A_n - 1)$,

$$\frac{1}{\sqrt{n}} E((n - L_1(1) - \ldots - L_1(A_n - 1)))(1 - \theta_{A_n}))$$

$$= \frac{1}{\sqrt{n}}(n - E(L_1(1) + \ldots + L_1(A_n - 1)))(1 - E(\theta_{A_n})),$$

which tends to $\frac{1}{\sqrt{2}}$ in view of (3) and (5). The announced bound follows. $\qquad\square$

## 4 Unknown time horizon

### 4.1 Anytime version of the algorithm

When the time horizon is unknown, the targets depend on the current time $t$, say $\ell_1(t)$ and $\ell_2(t)$. Now any arm that is exploited may be eventually discarded, in the sense that a new arm is explored. This happens whenever either $L_1 < \ell_1(t)$ or $L_2 < \ell_2(t)$, where $L_1$ and $L_2$ are the respective lengths of the first run and the first $m$ runs of this arm. Thus, unlike the previous version of the algorithm which consists in an exploration phase followed by an exploitation phase, the anytime version of the algorithm continuously switches between exploration and exploitation. We prove in Proposition 2 below that, for large $m$, the target values $\ell_1(t) = \lfloor \sqrt[3]{t} \rfloor$ and $\ell_2(t) = \lfloor m\sqrt{t} \rfloor$ given in the pseudo-code achieve an asymptotic regret in $2\sqrt{n}$.

---

**Algorithm 2:** Two-target algorithm with unknown time horizon.

---

**Parameter:** $m$
**Function:**
*Explore*
$I \leftarrow I + 1$, $L \leftarrow 0$, $M \leftarrow 0$
**Algorithm:**
$I \leftarrow 0$
*Explore*
Exploit $\leftarrow$ **false**
**forall the** $t = 1, 2, \ldots$ **do**
    Get reward $X$ from arm $I$
    $\ell_1 = \lfloor \sqrt[3]{t} \rfloor$, $\ell_2 = \lfloor m\sqrt{t} \rfloor$
    **if** Exploit **then**
        **if** $L_1 < \ell_1$ **or** $L_2 < \ell_2$ **then**
            *Explore*
            Exploit $\leftarrow$ **false**
    **else**
        **if** $X = 1$ **then**
            $L \leftarrow L + 1$
        **else**
            $M \leftarrow M + 1$
            **if** $M = 1$ **then**
                **if** $L < \ell_1$ **then**
                    *Explore*
                **else**
                    $L_1 \leftarrow L$
            **else if** $M = m$ **then**
                **if** $L < \ell_2$ **then**
                    *Explore*
                **else**
                  $L_2 \leftarrow L$
                  Exploit$\leftarrow$ **true**

---

## 4.2 Regret analysis

**Proposition 2** *The two-target algorithm with time-dependent targets $\ell_1(t) = \lfloor \sqrt[3]{t} \rfloor$ and $\ell_2(t) = \lfloor m\sqrt{t} \rfloor$ satisfies:*

$$\limsup_{n \to +\infty} \frac{E(R_n)}{\sqrt{n}} \le 2 + \frac{1}{m}.$$

*Proof.* For all $k = 1, 2, \ldots$, denote by $L_1(k)$ and $L_2(k)$ the respective lengths of the first run and of the first $m$ runs of arm $k$ when this arm is played continuously. Since arm $k$ cannot be selected before time $k$, the regret at time $n$ satisfies:

$$R_n \le K_n + m \sum_{k=1}^{K_n} 1_{\{L_1(k) > \ell_1(k)\}} + \sum_{t=1}^{n} (1 - X_t) 1_{\{L_2(I_t) > \ell_2(t)\}}.$$

First observe that, since the target functions $\ell_1(t)$ and $\ell_2(t)$ are non-decreasing, $K_n$ is less than or equal to $K_n'$, the number of arms selected by a two-target policy with known time horizon $n$ and fixed targets $\ell_1(n)$ and $\ell_2(n)$. In this scheme, let $U_1' = 1$ if arm 1 is used until time $n$ and $U_1' = 0$ otherwise. It then follows from (2) that $P(U_1' = 1) \sim \frac{1}{\sqrt{n}}$ and $E(K_n) \le E(K_n') \sim \sqrt{n}$ when $n \to +\infty$.

Now,

$$E\left( \sum_{k=1}^{K_n} 1_{\{L_1(k) > \ell_1(k)\}} \right) = \sum_{k=1}^{\infty} P(L_1(k) > \ell_1(k), K_n \ge k),$$

$$= \sum_{k=1}^{\infty} P(L_1(k) > \ell_1(k)) P(K_n \ge k | L_1(k) > \ell_1(k)),$$

$$\le \sum_{k=1}^{\infty} P(L_1(k) > \ell_1(k)) P(K_n \ge k) \le \sum_{k=1}^{E(K_n)} P(L_1(k) > \ell_1(k)),$$

where the first inequality follows from the fact that for any arm $k$ and all $u \in [0, 1]$,

$$P(\theta_k \ge u | L_1(k) > \ell_1(k)) \ge P(\theta_k \ge u) \quad \text{and} \quad P(K_n \ge k | \theta_k \ge u) \le P(K_n \ge k),$$

and the second inequality follows from the fact that the random variables $L_1(1), L_1(2), \ldots$ are i.i.d. and the sequence $\ell_1(1), \ell_1(2), \ldots$ is non-decreasing. Since $E(K_n) \le E(K_n') \sim \sqrt{n}$ when $n \to +\infty$ and $P(L_1(k) > \ell_1(k)) \sim \frac{1}{\sqrt[3]{k}}$ when $k \to +\infty$, we deduce:

$$\lim_{n \to +\infty} \frac{1}{\sqrt{n}} E\left( \sum_{k=1}^{K_n} 1_{\{L_1(k) > \ell_1(k)\}} \right) = 0.$$

Finally,

$$E((1 - X_t) 1_{\{L_2(I_t) > \ell_2(t)\}}) \le E(1 - X_t | L_2(I_t) > \ell_2(t)) \sim \frac{m+1}{m} \frac{1}{2\sqrt{t}} \quad \text{when } t \to +\infty,$$

so that

$$\limsup_{n \to +\infty} \frac{1}{\sqrt{n}} \sum_{t=1}^{n} E((1 - X_t) 1_{\{L_2(I_t) > \ell_2(t)\}}) \le \frac{m+1}{m} \lim_{n \to \infty} \frac{1}{n} \sum_{t=1}^{n} \frac{1}{2} \sqrt{\frac{n}{t}},$$

$$= \frac{m+1}{m} \int_0^1 \frac{1}{2\sqrt{u}} du = \frac{m+1}{m}.$$

Combining the previous results yields:

$$\limsup_{n \to +\infty} \frac{E(R_n)}{\sqrt{n}} \le 2 + \frac{1}{m}.$$

$\square$

### 4.3 Lower bound

We believe that if $E(R_n)/\sqrt{n}$ tends to some limit, then this limit is at least 2. To support this conjecture, consider an oracle that reveals the parameter of each arm after the first failure of this arm, as in the proof of Theorem 1. With this information, an optimal policy exploits an arm whenever its parameter is larger than some increasing function $\bar{\theta}_t$ of time $t$. Assume that $1 - \bar{\theta}_t \sim \frac{1}{c\sqrt{t}}$ for some $c > 0$ when $t \to +\infty$. Then proceeding as in the proof of Theorem 1, we get:

$$\liminf_{n \to +\infty} \frac{E(R_n)}{\sqrt{n}} \geq c + \lim_{n \to +\infty} \frac{1}{n} \sum_{t=1}^{n} \frac{1}{2c} \sqrt{\frac{n}{t}} = c + \frac{1}{c} \int_0^1 \frac{\mathrm{d}u}{2\sqrt{u}} = c + \frac{1}{c} \geq 2.$$

## 5   Numerical results

Figure 1 gives the expected failure rate $E(R_n)/n$ with respect to the time horizon $n$, that is supposed to be known. The results are derived from the simulation of $10^5$ independent samples and shown with 95% confidence intervals. The mean rewards have (a) a uniform distribution or (b) a Beta(1,2) distribution, corresponding to the probability density function $u \mapsto 2(1-u)$. The single-target algorithm corresponds to the run policy of Berry et. al. [2] with the asymptotically optimal target values $\sqrt{n}$ and $\sqrt[3]{2n}$, respectively. For the two-target algorithm, we take $m = 3$ and the target values given in Proposition 1 and Proposition 3 (in the appendix). The results are compared with the respective asymptotic lower bounds $\sqrt{2/n}$ and $\sqrt[3]{3/n}$. The performance gains of the two-target algorithm turn out to be negligible for the uniform distribution but substantial for the Beta(1,2) distribution, where "good" arms are less frequent.

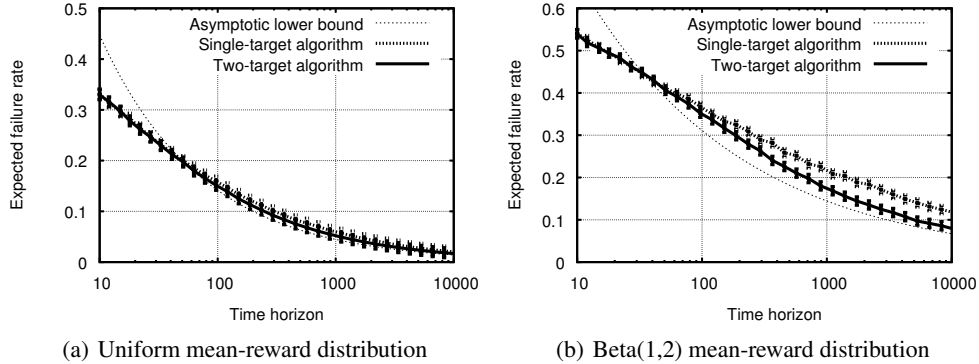

(a) Uniform mean-reward distribution          (b) Beta(1,2) mean-reward distribution

Figure 1: Expected failure rate $E(R_n)/n$ with respect to the time horizon $n$.

## 6   Conclusion

The proposed algorithm uses two levels of sampling in the exploration phase: the first eliminates "bad" arms while the second selects "good" arms. To our knowledge, this is the first algorithm that achieves the optimal regrets in $\sqrt{2n}$ and $2\sqrt{n}$ for known and unknown horizon times, respectively. Future work will be devoted to the proof of the lower bound in the case of unknown horizon time. We also plan to study various extensions of the present work, including mean-reward distributions whose support does not contain 1 and distribution-free algorithms. Finally, we would like to compare the performance of our algorithm for finite-armed bandits with those of the best known algorithms like KL-UCB [10, 3] and Thompson sampling [14, 8] over short time horizons where the full exploration of the arms is generally not optimal.

**Acknowledgments**

The authors acknowledge the support of the European Research Council, of the French ANR (GAP project), of the Swedish Research Council and of the Swedish SSF.

## Footnotes

[1]Specifically, it is assumed that for any policy, the mean rewards of the explored arms have a uniform distribution over $[0, 1]$, independently of the number of explored arms. This is incorrect. For the 1-failure policy for instance, given that only one arm has been explored until time $n$, the mean reward of this arm has a beta distribution with parameters $1, n$.

[2]This lower bound is $4\sqrt{n/3}$ for a beta distribution with parameters $1/2, 1$, see [11], while our algorithm achieves a regret arbitrarily close to $2\sqrt{n}$ in this case, since $C(\alpha, \beta) = 2$ for $\alpha = 1/2$ and $\beta = 1$, see the appendix. Thus the statement of [2, Theorem 11] is false.

[3]The first target could actually be any function $\ell_1$ of the time horizon $n$ such that $\ell_1 \to +\infty$ and $\ell_1/\sqrt{n} \to 0$ when $n \to +\infty$. Only the second target is critical.

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
