[Supplementary Material]

# Two-Target Algorithms for Infinite-Armed Bandits with Bernoulli Rewards – Appendix

**Thomas Bonald**[*]
Department of Networking and Computer Science
Telecom ParisTech
Paris, France
thomas.bonald@telecom-paristech.fr

**Alexandre Proutière**[*][†]
Automatic Control Department
KTH
Stockholm, Sweden
alepro@kth.se

In this appendix, we extend the results to any mean-reward distribution whose support contains 1. We denote by $F$ the complementary cumulative distribution function, i.e., $F(u) = P(\theta_1 > u)$ for all $u \in [0, 1]$. We assume that $F(1) = 0$ and $F(u) \sim \alpha(1 - u)^\beta$ when $u \to 1^-$ for some constants $\alpha, \beta > 0$.

## A  Preliminary results

We need the following two technical lemmas, where $\Gamma$ refers to the gamma function:

**Lemma 1** *For all $\beta > 0$ and $j = 0, 1, \ldots$,*

$$\sum_{i=0}^{j} \binom{j}{i} \frac{(-1)^i}{(1 - (j-i)x)^\beta} \sim_{x \to 0} \frac{\Gamma(\beta + j)}{\Gamma(\beta)} x^j.$$

*Proof.* Let:

$$f_j(x) = \sum_{i=0}^{j} \binom{j}{i} \frac{(-1)^i}{(1 - (j-i)x)^\beta}.$$

For all $k = 1, 2, \ldots$, the $k$-th derivative of $f_j$ in $x = 0$ is given by:

$$f_j^{(k)}(0) = \frac{\Gamma(\beta + k)}{\Gamma(\beta)} \sum_{i=0}^{j-1} \binom{j}{i} (-1)^i (j-i)^k.$$

We need to prove that $f_j^{(k)}(0) = 0$ for all $k < j$ and:

$$f_j^{(j)}(0) = \frac{\Gamma(\beta + j)}{\Gamma(\beta)} j!,$$

with the convention that $f_j^{(0)} \equiv f_j$. We prove the result by induction on $j$. The property holds for $j = 0$. Assume that the property is satisfied for $j - 1$, for some $j \geq 1$. Note that $f_j(0) = 0$. Now for all $k = 1, 2, \ldots$,

$$
\begin{aligned}
f_j^{(k)}(0) \frac{\Gamma(\beta)}{\Gamma(\beta + k)} &= \sum_{i=0}^{j-1} \binom{j}{i} (-1)^i (j-i)^k, \\
&= j \sum_{i=0}^{j-1} \binom{j-1}{i-1} (-1)^i (j-i)^{k-1},
\end{aligned}
$$

which, by the induction assumption, is equal to 0 if $k < j$ and to $j!$ if $k = j$. $\square$

---

[*]The authors are members of the LINCS, Paris, France. See www.lincs.fr.

[†]Alexandre Proutière is also affiliated to INRIA, Paris-Rocquencourt, France.

**Lemma 2** *For all $\beta > 0$,*

$$\lim_{m \to +\infty} \frac{1}{m^\beta} \sum_{j=0}^{m-1} \frac{\Gamma(\beta + j)}{j!} = \frac{1}{\beta}.$$

*Proof.* Since

$$\frac{\Gamma(\beta + j)}{j!} \sim j^{\beta - 1} \quad \text{when } j \to +\infty,$$

we have:

$$\lim_{m \to +\infty} \frac{1}{m^\beta} \sum_{j=0}^{m-1} \frac{\Gamma(\beta + j)}{j!} = \lim_{m \to +\infty} \frac{1}{m} \sum_{j=0}^{m-1} \left(\frac{j}{m}\right)^{\beta - 1} = \int_0^1 u^{\beta - 1} du = \frac{1}{\beta}.$$

$\square$

# B   Known horizon time

## B.1   Regret analysis

**Proposition 3** *The two-target algorithm with targets $\ell_1 = \left\lfloor \left(\frac{\alpha n}{\beta + 1}\right)^{\frac{1}{\beta + 2}} \right\rfloor$ and $\ell_2 = \left\lfloor m \left(\frac{\alpha n}{\beta + 1}\right)^{\frac{1}{\beta + 1}} \right\rfloor$ satisfies:*

$$\lim_{m \to +\infty} \limsup_{n \to +\infty} \frac{E(R_n)}{n^{\frac{\beta}{\beta + 1}}} \leq \left(\frac{\beta + 1}{\alpha}\right)^{\frac{1}{\beta + 1}}.$$

*Proof.* Let $U_1 = 1$ if arm 1 is used until time $n$ and $U_1 = 0$ otherwise. Denote by $M_1$ the total number of 0's received from arm 1:

$$M_1 = \sum_{t=1}^{n} 1_{\{I_t = 1\}} 1_{\{X_t = 0\}}.$$

We have:

$$E(R_n) \leq P(U_1 = 0)(E(M_1 | U_1 = 0) + E(R_n)) + P(U_1 = 1)(m + nE(1 - \theta_1 | U_1 = 1)),$$

so that:

$$E(R_n) \leq \frac{E(M_1 | U_1 = 0)}{P(U_1 = 1)} + m + nE(1 - \theta_1 | U_1 = 1).$$

Let $N_t$ be the number of 0's received from arm 1 until time $t$ when this arm is played until time $t$. We take $n$ sufficiently large so that $n \geq \ell_2$. Since $P(N_{\ell_1} = 0 | \theta_1 = u) = u^{\ell_1}$, the probability that the first target is achieved by arm 1 is given by:

$$
\begin{aligned}
P(N_{\ell_1} = 0) \quad &= \quad E(\theta_1^{\ell_1}), \\
&= \quad \int_0^1 F(u^{\frac{1}{\ell_1}}) du, \\
&\sim_{n \to +\infty} \quad \int_0^1 \alpha \left(\frac{-\ln(u)}{\ell_1}\right)^\beta du = \alpha \frac{\Gamma(\beta + 1)}{\ell_1^\beta},
\end{aligned}
$$

where we have used the fact that:

$$\forall x > 0, \quad \Gamma(x) \equiv \int_0^{+\infty} t^{x-1} e^{-t} dt = \int_0^1 (-\ln(u))^{x-1} du.$$

For the second target, we have:

$$P(N_{\ell_2 - \ell_1} < m | \theta_1 = u) = \sum_{j=0}^{m-1} \binom{\ell_2 - \ell_1}{j} u^{\ell_2 - \ell_1 - j} (1 - u)^j,$$

so that:

$$
\begin{aligned}
P(U_1 = 1|\theta = u) &= P(N_{\ell_1} = 0, N_{\ell_2 - \ell_1} < m|\theta_1 = u), \\
&= \sum_{j=0}^{m-1} \binom{\ell_2 - \ell_1}{j} u^{\ell_2 - j}(1 - u)^j.
\end{aligned}
\tag{1}
$$

We deduce the probability that arm 1 is used until time $n$:

$$
P(U_1 = 1) = \sum_{j=0}^{m-1} \binom{\ell_2 - \ell_1}{j} E(\theta_1^{\ell_2 - j}(1 - \theta_1)^j).
$$

For all $j = 0, \ldots, m - 1$, we have:

$$
\begin{aligned}
E(\theta_1^{\ell_2 - j}(1 - \theta_1)^j) &= \sum_{i=0}^{j} \binom{j}{i}(-1)^i E(\theta_1^{\ell_2 - j + i}), \\
&\sim_{n \to +\infty} \sum_{i=0}^{j} \binom{j}{i}(-1)^i \alpha \frac{\Gamma(\beta + 1)}{(\ell_2 - j + i)^\beta}, \\
&\sim_{n \to +\infty} \alpha\beta \frac{\Gamma(\beta + j)}{\ell_2^{\beta + j}},
\end{aligned}
$$

where the last equivalent follows from Lemma 1. We deduce:

$$
\begin{aligned}
P(U_1 = 1) &\sim_{n \to +\infty} \sum_{j=0}^{m-1} \frac{\ell_2^j}{j!} \alpha\beta \frac{\Gamma(\beta + j)}{\ell_2^{\beta + j}}, \\
&\sim_{n \to +\infty} \alpha\beta\gamma(m) \left(\frac{\beta + 1}{\alpha n}\right)^{\frac{\beta}{\beta + 1}},
\end{aligned}
$$

with

$$
\gamma(m) = \frac{1}{m^\beta} \sum_{j=0}^{m-1} \frac{\Gamma(\beta + j)}{j!}.
$$

Now,

$$
E(M_1|U_1 = 0) = 1 + (m - 1)P(N_{\ell_1} = 0|U_1 = 0) \leq 1 + (m - 1)\frac{P(N_{\ell_1} = 0)}{P(U_1 = 0)}.
$$

Since $P(N_{\ell_1} = 0)$ and $P(U_1 = 0)$ tend respectively to 0 and 1 when $n \to +\infty$, we deduce:

$$
\limsup_{n \to +\infty} E(M_1|U_1 = 0) \leq 1.
$$

It remains to calculate $E(1 - \theta_1|U_1 = 1)$. Using (1), we get:

$$
\begin{aligned}
E(1 - \theta_1|U_1 = 1) &= \frac{E((1 - \theta_1)1_{\{U_1 = 1\}})}{P(U_1 = 1)}, \\
&= \frac{1}{P(U_1 = 1)} \sum_{j=0}^{m-1} \binom{\ell_2 - \ell_1}{j} E(\theta_1^{\ell_2 - j}(1 - \theta_1)^{j+1}).
\end{aligned}
$$

Since

$$
E(\theta_1^{\ell_2 - j}(1 - \theta_1)^{j+1}) \sim_{n \to +\infty} \alpha\beta \frac{\Gamma(\beta + j + 1)}{\ell_2^{\beta + j + 1}},
$$

we obtain:

$$
\begin{aligned}
\sum_{j=0}^{m-1} \binom{\ell_2 - \ell_1}{j} E(\theta_1^{\ell_2 - j}(1 - \theta_1)^{j+1}) &\sim_{n \to +\infty} \sum_{j=0}^{m-1} \frac{\ell_2^j}{j!} \alpha\beta \frac{\Gamma(\beta + j + 1)}{\ell_2^{\beta + j + 1}}, \\
&\sim_{n \to +\infty} \alpha\beta\gamma'(m)\frac{\beta + 1}{\alpha n},
\end{aligned}
$$

with

$$\gamma'(m) = \frac{1}{m^{\beta+1}} \sum_{j=0}^{m-1} \frac{\Gamma(\beta+j+1)}{j!},$$

and

$$E(1 - \theta_1 | U_1 = 1) \sim_{n \to +\infty} \frac{\gamma'(m)}{\gamma(m)} \left(\frac{\beta+1}{\alpha n}\right)^{\frac{1}{\beta+1}}.$$

Finally, we get:

$$\limsup_{n \to +\infty} \frac{E(R_n)}{n^{\frac{\beta}{\beta+1}}} \le \frac{1}{\alpha\beta\gamma(m)} \left(\frac{\alpha}{\beta+1}\right)^{\frac{\beta}{\beta+1}} + \frac{\gamma'(m)}{\gamma(m)} \left(\frac{\beta+1}{\alpha}\right)^{\frac{1}{\beta+1}},$$

and the result follows from the fact that, by Lemma 2, $\gamma(m) \to \frac{1}{\beta}$ and $\gamma'(m) \to \frac{1}{\beta+1}$ when $m \to +\infty$. $\qquad\square$

## B.2   Lower bound

To prove the lower bound, we need the following technical result:

**Lemma 3** *Let $L$ be the length of the first run of any arm, with parameter $\theta$. For any sequence of positive numbers $\epsilon_k = O(k^{-\frac{1}{\beta+1}})$, we have:*

$$E(L | \theta \le 1 - \epsilon_k) = \begin{cases} O(k^{\frac{1-\beta}{\beta+1}}) & \text{if } \beta < 1, \\ O(\ln k) & \text{if } \beta = 1, \\ O(1) & \text{if } \beta > 1. \end{cases}$$

*Moreover, for any sequence of positive numbers $\epsilon_k$ tending to 0,*

$$\lim_{k \to +\infty} \epsilon_k E(L | \theta \le 1 - \epsilon_k) = 0.$$

*Proof.* We have:

$$E(L | \theta \le 1 - \epsilon_k) = E\left(\frac{1}{1-\theta} \Big| \theta \le 1 - \epsilon_k\right).$$

Let $c < 1$ be such that $F(u) \le 2\alpha(1-u)^\beta$ for all $u > c$. Take $k$ sufficiently large so that $1 - \epsilon_k > c$. For some constant $C > 0$,

$$\begin{aligned}
E(L | \theta \le 1 - \epsilon_k) &= 1 + \frac{\int_1^{\frac{1}{\epsilon_k}} \left(F(1 - \frac{1}{u}) - F(1 - \epsilon_k)\right) \mathrm{d}u}{1 - F(1 - \epsilon_k)}, \\
&\le C + \frac{\int_{\frac{1}{1-c}}^{\frac{1}{\epsilon_k}} F(1 - \frac{1}{u}) \mathrm{d}u}{1 - F(1 - \epsilon_k)}, \\
&\le C + \frac{\int_{\frac{1}{1-c}}^{\frac{1}{\epsilon_k}} \frac{2\alpha}{u^\beta} \mathrm{d}u}{1 - F(1 - \epsilon_k)}.
\end{aligned}$$

We distinguish between three cases:

- If $\beta < 1$,

$$\int_{\frac{1}{1-c}}^{\frac{1}{\epsilon_k}} \frac{\mathrm{d}u}{u^\beta} = \frac{\epsilon_k^{\beta-1} - (1-c)^{\beta-1}}{1-\beta} \le \frac{\epsilon_k^{\beta-1}}{1-\beta}.$$

- If $\beta = 1$,

$$\int_{\frac{1}{1-c}}^{\frac{1}{\epsilon_k}} \frac{\mathrm{d}u}{u^\beta} = -\ln(\epsilon_k) + \ln(1-c) \le -\ln(\epsilon_k).$$

- If $\beta > 1$,

$$\int_{\frac{1}{1-c}}^{\frac{1}{\epsilon_k}} \frac{\mathrm{d}u}{u^\beta} = \frac{\epsilon_k^{\beta-1} - (1-c)^{\beta-1}}{1-\beta} \leq \frac{(1-c)^{\beta-1}}{\beta-1}.$$

The proof then follows from the fact that $F(1 - \epsilon_k) \to 0$ when $k \to +\infty$. $\qquad\square$

**Theorem 2** *For any algorithm with known time horizon $n$,*

$$\liminf_{n \to +\infty} \frac{E(R_n)}{n^{\frac{\beta}{\beta+1}}} \geq \left(\frac{\beta+1}{\alpha}\right)^{\frac{1}{\beta+1}}.$$

*Proof.* Assume an oracle reveals the parameter of each arm after the first failure of this arm. With this information, the optimal policy explores a random number of arms, each until the first failure, then plays only one of these arms until time $n$. Let $\mu$ be the parameter of the best known arm at time $t$. We need to characterize the optimal policy for the remaining $k = n - t$ time steps. Let $V_1(k, \mu) = k(1 - \mu)$ be the expected number of failures when the best known arm is exploited for the remaining $k$ time steps and $V_2(k, \mu)$ be the expected number of failures when exactly one additional arm, with random parameter $\theta$, is explored:

$$V_2(k, \mu) = E\left(\sum_{t=0}^{k-1} \theta^t (1 - \theta)\left(1 + V_1(k - t - 1, \mu \vee \theta)\right)\right).$$

Now let $V(k, \mu)$ be the minimum expected number of failures when $k$ steps remain and $\mu$ is the parameter of the best known arm. This is the value function of a Markov Decision process, which satisfies Bellman's equation:

$$V(k, \mu) = \min\left(V_1(k, \mu), E\left(\sum_{t=0}^{k-1} \theta^t (1 - \theta)\left(1 + V(k - t - 1, \mu \vee \theta)\right)\right)\right). \qquad (2)$$

By construction, we have, for any algorithm, $E(R_n) \geq V(n, 0)$ for all $n \geq 1$.

Observe that for all $k \geq 1$, there exists some threshold $\mu_k$ such that $V_1(k, \mu) \leq V_2(k, \mu)$ if and only if $\mu \geq \mu_k$. This follows from the fact that $V_2(k, \mu) - V_1(k, \mu)$ is a non-decreasing function of $\mu$:

$$\frac{\partial V_2}{\partial \mu}(k, \mu) - \frac{\partial V_1}{\partial \mu}(k, \mu) = -E\left(\sum_{t=0}^{k-1} \theta^t (1 - \theta)(k - t - 1)1_{\{\theta \leq \mu\}}\right) + k \geq 0.$$

Now we study the behaviour of the threshold $\mu_k$ for large $k$. Define $\epsilon_k$ by:

$$\int_{1-\epsilon_k}^1 F(u)\mathrm{d}u = \frac{1}{k}.$$

Since

$$\int_{1-\epsilon}^1 F(u)\mathrm{d}u \sim \frac{\alpha}{\beta+1}\epsilon^{\beta+1} \quad \text{when } \epsilon \to 0^+,$$

we get:

$$\epsilon_k \sim \left(\frac{\beta+1}{\alpha k}\right)^{\frac{1}{\beta+1}} \quad \text{when } k \to +\infty.$$

We shall prove that $\mu_k \geq 1 - \epsilon_k$ and $1 - \mu_k \sim \epsilon_k$ when $k \to +\infty$. To do so, we compare the average rewards obtained from state $(k, \mu)$ if we do not explore any new arm and if we explore exactly one more arm. We divide the $k$ remaining rounds into the exploration phase, where the new arm is played until the first failure, and the exploitation phase where the best arm is played. Given the parameter $\theta$ of the new arm and the length $L$ of its first run, the expected number of failures removed thanks to the exploration of the new arm is:

$$L(1 - \mu) - 1 + (k - L)(\mu \vee \theta - \mu) = k(\mu \vee \theta - \mu) - 1 + L(1 - \mu \vee \theta).$$

Taking expectation, we conclude that it is beneficial to explore a new arm if and only if:

$$k\int_\mu^1 F(u)\mathrm{d}u - 1 + E(L(1 - \mu \vee \theta)) \geq 0.$$

In particular, it is better to explore a new arm whenever $\mu \leq 1 - \epsilon_k$ and thus $\mu_k \geq 1 - \epsilon_k$. Now let $\delta \in (0, 1)$ and assume that $\mu > 1 - \epsilon_k(1 - \delta)$. Since

$$k \int_{1-\epsilon_k}^{1-\epsilon_k(1-\delta)} F(u)\mathrm{d}u \geq k\epsilon_k \delta F(1 - \epsilon_k(1-\delta)) \sim_{k\to+\infty} \delta(1-\delta)^\beta,$$

there exists some constant $c > 0$ such that for sufficiently large $k$,

$$k \int_\mu^1 F(u)\mathrm{d}u - 1 \leq -c.$$

Moreover,

$$E(L(1 - \mu \vee \theta)) \leq P(\theta > \mu) + (1 - \mu)E(L|\theta \leq \mu),$$

which tends to 0 in view of Lemma 3. Thus, for sufficiently large $k$, it is better not to explore a new arm and thus $\mu_k \leq 1 - \epsilon_k(1 - \delta)$. We have proved that for any $\delta \in (0, 1)$, there exists $k_0$ such that for all $k \geq k_0$, $1 - \epsilon_k \leq \mu_k \leq 1 - \epsilon_k(1 - \delta)$.

Next we show that $\mu_k$ is non-decreasing in $k$ for $k$ large enough. To prove this, we establish that, for $k$ large enough,

$$V_2(k, \mu_k) - V_2(k - 1, \mu_k) \leq 1 - \mu_k. \tag{3}$$

This implies:

$$V_1(k - 1, \mu_k) = V_1(k, \mu_k) - (1 - \mu_k) = V_2(k, \mu_k) - (1 - \mu_k) \leq V_2(k - 1, \mu_k),$$

and thus $\mu_k \geq \mu_{k-1}$. Since

$$V_2(k, \mu) - V_2(k - 1, \mu) = E(1_{L<k-1}(1 - \theta \vee \mu) + 1_{L=k-1}),$$
$$= E((1 - \theta \vee \mu)(1 - \theta^{k-1}) + \theta^{k-1}(1 - \theta)),$$

inequality (3) is equivalent to:

$$E(\mu_k - \theta \vee \mu_k - \theta^{k-1}(\theta - \theta \vee \mu_k)) \leq 0.$$

Now, we can choose $\delta \in (0, 1)$ and hence $k_0$ such that for all $k \geq k_0$:

$$E(\theta \vee \mu_k - \mu_k) = \int_{\mu_k}^1 F(u)du \geq \int_{1-\epsilon_k(1-\delta)}^1 F(u)du \geq \frac{1}{2}\int_{1-\epsilon_k}^1 F(u)du = \frac{1}{2k}.$$

Moreover, for all $k \geq k_0$,

$$E(\theta^{k-1}(\theta \vee \mu_k - \theta)) \leq \mu_k^{k-1} E(1_{\theta \leq \mu_k}(\mu_k - \theta)) \leq \mu_k^k \leq (1 - \epsilon_k(1 - \delta))^k.$$

Using the fact that $\epsilon_k \sim \left(\frac{\beta+1}{\alpha k}\right)^{\frac{1}{\beta+1}}$, we deduce that there exists some $k_1 \geq k_0$ such that for all $k \geq k_1$,

$$(1 - \epsilon_k(1 - \delta))^k \leq \frac{1}{2k},$$

which proves that (3) holds for $k \geq k_1$.

Finally, we compute $V(k, \mu)$. Specifically, we prove that for $k$ large enough, $V(k, \mu) = V_1(k, \mu)$ if and only if $\mu \geq \mu_k$. If $\mu < \mu_k$, then

$$V(k, \mu) \leq V_2(k, \mu) < V_1(k, \mu).$$

Since $\mu_k$ is arbitrarily close to 1 as $k \to +\infty$, there exists $k_2 \geq k_1$ such that for all $k \geq k_2$, $\mu_k \geq \max(\mu_1, \ldots, \mu_{k_1})$. Hence, $\mu_k \geq \mu_s$ for all $s \leq k$, using the monotonicity of $\mu_k$ for $k \geq k_1$. Now assume that $\mu \geq \mu_k$. We prove by induction on $t \in \{1, \ldots, k\}$ that $V(t, \lambda) = V_1(t, \lambda)$ for all $\lambda \geq \mu$. For $t = 1$, this is immediate since $\lambda \geq \mu \geq \mu_k \geq \mu_1$. Assume that the property holds for all $t \leq s$, for some $s < k$, and let us prove it for $s + 1$. Since $\lambda \geq \mu_k \geq \mu_{s+1}$,

$$V_1(s + 1, \lambda) \leq V_2(s + 1, \lambda) = E\left(\sum_{t=0}^s \theta^t(1 - \theta)(1 + V_1(s - t, \lambda \vee \theta))\right),$$

$$= E\left(\sum_{t=0}^s \theta^t(1 - \theta)(1 + V(s - t, \lambda \vee \theta))\right),$$

where the last equality is obtained by the induction assumption, since $\lambda \geq \mu$. It then follows from Bellman's equation (2) that $V(s+1, \lambda) = V_1(s+1, \lambda)$.

From the above analysis, we know that if there remain $k$ timesteps, for some $k \geq k_2$, then a new arm must be explored if and only if $\mu \geq \mu_k$. To conclude the proof, let $\varepsilon > 0$ and denote by $A_n$ the first arm whose parameter $\theta$ satisfies:

$$\frac{\alpha}{\beta+1}(1-\theta)^{\beta+1} < \frac{1-\varepsilon}{n}$$

and by $\bar{\theta}_n$ the corresponding limiting parameter:

$$(1-\bar{\theta}_n)^{\beta+1} = \frac{(1-\varepsilon)(\beta+1)}{\alpha n}.$$

Since

$$\int_{\bar{\theta}_n}^1 F(u)\mathrm{d}u \sim \frac{1-\varepsilon}{n} \quad \text{when } n \to +\infty,$$

we have $\bar{\theta}_n \geq 1 - \epsilon_n$ and thus the number of explored arms $K_n$ satisfies $K_n \leq A_n$ for sufficiently large $n$. Moreover,

$$E(A_n) = \frac{1}{F(\bar{\theta}_n)} \sim \frac{1}{\alpha}\left(\frac{\alpha n}{(1-\varepsilon)(\beta+1)}\right)^{\frac{\beta}{\beta+1}} \quad \text{when } n \to +\infty.$$

The parameter $\theta_{A_n}$ of arm $A_n$ is independent of $A_n$ and satisfies:

$$
\begin{aligned}
1 - E(\theta_{A_n}) &= 1 - E(\theta|\theta > \bar{\theta}_n), \\
&= 1 - \bar{\theta}_n - \frac{\int_{\bar{\theta}_n}^1 F(u)\mathrm{d}u}{F(\bar{\theta}_n)}, \\
&\sim \frac{\beta}{\beta+1}(1-\bar{\theta}_n) = \frac{\beta}{\beta+1}\left(\frac{(1-\varepsilon)(\beta+1)}{\alpha n}\right)^{\frac{1}{\beta+1}} \quad \text{when } n \to +\infty.
\end{aligned}
$$

For all $k = 1, 2, \ldots$, let $L_1(k)$ be the length of the first run of arm $k$ (until the first failure):

$$
\begin{aligned}
E(L_1(1) + \ldots + L_1(A_n - 1)) &= \sum_{k=1}^{\infty} P(A_n = k)E(L_1(1) + \ldots + L_1(k-1)|A_n = k), \\
&= \sum_{k=1}^{\infty} P(A_n = k)E(L_1(1) + \ldots + L_1(k-1)|\theta_1, \ldots, \theta_{k-1} \leq \bar{\theta}_n), \\
&= \sum_{k=1}^{\infty} P(A_n = k)(k-1)E(L_1(1)|\theta_1 \leq \bar{\theta}_n), \\
&= (E(A_n) - 1)E(L_1(1)|\theta_1 \leq \bar{\theta}_n).
\end{aligned}
$$

Using Lemma 3, we get:

$$E(L_1(1) + \ldots + L_1(A_n - 1)) = \begin{cases} O(n^{\frac{1}{\beta+1}}) & \text{if } \beta < 1, \\ O(\sqrt{n}\ln(n)) & \text{if } \beta = 1, \\ O(n^{\frac{\beta}{\beta+1}}) & \text{if } \beta > 1. \end{cases}$$

Let:

$$\varphi(\beta) = \begin{cases} \frac{1}{1+\beta/2} & \text{if } \beta \leq 1, \\ \frac{\beta}{\beta+1/2} & \text{if } \beta > 1, \end{cases} \quad \text{and} \quad \phi(\beta) = \begin{cases} \frac{1}{1+\beta/3} & \text{if } \beta \leq 1, \\ \frac{\beta}{\beta+1/3} & \text{if } \beta > 1, \end{cases}$$

Observe that $\varphi(\beta) < \phi(\beta) < 1$. We have

$$E(L_1(1) + \ldots + L_1(A_n - 1))/n^{\varphi(\beta)} \to 0$$

and

$$P(L_1(1) + \ldots + L_1(A_n - 1) \leq n^{\phi(\beta)}) \to 1$$

when $n \to +\infty$. For the latter, assume that $c \equiv \liminf_n P(L_1(1) + \ldots + L_1(A_n - 1) > n^{\phi(\beta)}) > 0$. Then,

$$\frac{E(L_1(1) + \ldots + L_1(A_n - 1))}{n^{\varphi(\beta)}} \geq c n^{\phi(\beta) - \varphi(\beta)} \to +\infty,$$

a contradiction.

Since $K_n \leq A_n$, we have:

$$E(R_n) \geq E(K_n) + E((n - L_1(1) - \ldots - L_1(A_n - 1))(1 - \theta_{A_n})).$$

Observe that, on the event $\{L_1(1) + \ldots + L_1(A_n - 1) \leq n^{\phi(\beta)}\}$, for sufficiently large $n$, the number of explored arms satisfies $K_n \geq A'_n$ where $A'_n$ denotes the first arm whose parameter $\theta$ is such that

$$\int_\theta^1 F(u)\mathrm{d}u < \frac{1}{n - n^{\phi(\beta)}}.$$

Since $P(L_1(1) + \ldots + L_1(A_n - 1) \leq n^{\phi(\beta)}) \to 1$ and

$$E(A'_n) \sim \frac{1}{\alpha}\left(\frac{\alpha(n - n^{\phi(\beta)})}{(1 - \varepsilon)(\beta + 1)}\right)^{\frac{\beta}{\beta+1}} \quad \text{when } n \to +\infty,$$

we get:

$$\liminf_{n \to +\infty} \frac{E(K_n)}{n^{\frac{\beta}{\beta+1}}} \geq \frac{1}{\alpha}\left(\frac{\alpha}{(1 - \varepsilon)(\beta + 1)}\right)^{\frac{\beta}{\beta+1}}.$$

By the independence of $\theta_{A_n}$ and $L_1(1), \ldots, L_1(A_n - 1)$,

$$\frac{E((n - L_1(1) - \ldots - L_1(A_n - 1))(1 - \theta_{A_n}))}{n^{\frac{\beta}{\beta+1}}}$$

$$= \frac{(n - E(L_1(1) + \ldots + L_1(A_n - 1)))(1 - E(\theta_{A_n}))}{n^{\frac{\beta}{\beta+1}}},$$

$$\to \frac{\beta}{\beta + 1}\left(\frac{(1 - \varepsilon)(\beta + 1)}{\alpha}\right)^{\frac{1}{\beta+1}}.$$

Letting $\varepsilon$ tend to 0 gives the desired bound:

$$\liminf_{n \to +\infty} \frac{E(R_n)}{n^{\frac{\beta}{\beta+1}}} \geq \frac{1}{\alpha}\left(\frac{\alpha}{\beta + 1}\right)^{\frac{\beta}{\beta+1}} + \frac{\beta}{\beta + 1}\left(\frac{\beta + 1}{\alpha}\right)^{\frac{1}{\beta+1}} = \left(\frac{\beta + 1}{\alpha}\right)^{\frac{1}{\beta+1}}.$$

$\qquad\qquad\qquad\qquad\qquad\qquad\qquad\qquad\qquad\qquad\qquad\qquad\qquad\qquad\qquad\qquad\qquad\qquad\qquad\square$

## C    Unkown horizon time

### C.1    Regret analysis

**Proposition 4** *The two-target algorithm with time-dependent targets* $\ell_1(t) = \left\lfloor \left(\frac{\alpha t}{\beta}\right)^{\frac{1}{\beta+2}} \right\rfloor$ *and* $\ell_2(t) = \left\lfloor m\left(\frac{\alpha t}{\beta}\right)^{\frac{1}{\beta+1}} \right\rfloor$ *satisfies:*

$$\lim_{m \to +\infty} \limsup_{n \to +\infty} \frac{E(R_n)}{n^{\frac{\beta}{\beta+1}}} \leq \frac{\beta + 1}{\beta}\left(\frac{\beta}{\alpha}\right)^{\frac{1}{\beta+1}}.$$

*Proof.* The proof is similar to that of Proposition 2. We have:

$$R_n \leq K_n + m\sum_{k=1}^{K_n} 1_{\{L_1(k) > \ell_1(k)\}} + \sum_{t=1}^{n}(1 - X_t)1_{\{L_2(I_t) > \ell_2(t)\}}.$$

Let $U_1' = 1$ if arm 1 is used until time $n$ and $U_1' = 0$ otherwise, for a two-target algorithm with known time horizon $n$ and targets $\ell_1(n)$ and $\ell_2(n)$. Denote by $K_n'$ the number of explored arms for this algorithm. By Lemma 1, we get as in the proof of Proposition 3:

$$P(U_1' = 1) \sim \alpha\beta\gamma(m) \left(\frac{\beta}{\alpha n}\right)^{\frac{\beta}{\beta+1}} \qquad \text{when } n \to +\infty,$$

so that:

$$E(K_n) \leq E(K_n') \sim \frac{1}{\alpha\beta\gamma(m)} \left(\frac{\alpha n}{\beta}\right)^{\frac{\beta}{\beta+1}} \qquad \text{when } n \to +\infty.$$

Moreover, as in the proof of Proposition 3,

$$\lim_{n \to +\infty} \frac{1}{n^{\frac{\beta}{\beta+1}}} E\left(\sum_{k=1}^{K_n} 1_{\{L_1(k) > \ell_1(k)\}}\right) = 0,$$

and

$$E((1 - X_t)1_{\{L_2(I_t) > \ell_2(t)\}}) \leq E(1 - X_t | L_2(I_t) > \ell_2(t)) \sim \frac{\gamma'(m)}{\gamma(m)} \left(\frac{\beta}{\alpha t}\right)^{\frac{1}{\beta+1}} \qquad \text{when } t \to +\infty,$$

so that:

$$\limsup_{n \to +\infty} \frac{1}{n^{\frac{\beta}{\beta+1}}} \sum_{t=1}^{n} E((1 - X_t)1_{\{L_2(I_t) > \ell_2(t)\}}) \leq \frac{\gamma'(m)}{\gamma(m)} \left(\frac{\beta}{\alpha}\right)^{\frac{1}{\beta+1}} \lim_{n \to +\infty} \frac{1}{n} \sum_{t=1}^{n} \left(\frac{n}{t}\right)^{\frac{1}{\beta+1}},$$

$$= \frac{\gamma'(m)}{\gamma(m)} \left(\frac{\beta}{\alpha}\right)^{\frac{1}{\beta+1}} \int_0^1 \frac{du}{u^{\frac{1}{\beta+1}}},$$

$$= \frac{\gamma'(m)}{\gamma(m)} \left(\frac{\beta}{\alpha}\right)^{\frac{1}{\beta+1}} \frac{\beta+1}{\beta}.$$

Combining the previous results yields:

$$\limsup_{n \to +\infty} \frac{E(R_n)}{n^{\frac{\beta}{\beta+1}}} \leq \frac{1}{\alpha\beta\gamma(m)} \left(\frac{\alpha}{\beta}\right)^{\frac{\beta}{\beta+1}} + \frac{\gamma'(m)}{\gamma(m)} \left(\frac{\beta}{\alpha}\right)^{\frac{1}{\beta+1}} \frac{\beta+1}{\beta},$$

$$= \frac{1}{\gamma(m)\beta} \left(\frac{\beta}{\alpha}\right)^{\frac{1}{\beta+1}} \left(\frac{1}{\beta} + \gamma'(m)(\beta+1)\right).$$

The proof follows from the fact that, by Lemma 2, $\beta\gamma(m) \to 1$ and $(\beta + 1)\gamma'(m) \to 1$ when $m \to +\infty$. $\qquad \square$

## C.2 Lower bound

As for the uniform mean-reward distribution, we conjecture that the two-target algorithm is optimal in the sense that, for any algorithm, if $E(R_n)/n^{\frac{\beta}{\beta+1}}$ tends to some limit, then this limit is at least $\frac{\beta+1}{\beta} \left(\frac{\beta}{\alpha}\right)^{\frac{1}{\beta+1}}$. Consider an oracle that reveals the parameter of each arm after the first failure of this arm, as in the proof of Theorem 2. With this information, an optimal policy exploits an arm whenever its parameter is larger than some increasing function $\bar{\theta}_t$ of time $t$.

Assume that $1 - \bar{\theta}_t \sim \frac{1}{ct^{\frac{1}{\beta+1}}}$ for some $c > 0$ when $t \to +\infty$. Then proceeding as in the proof of Theorem 2, we get:

$$\liminf_{n \to +\infty} \frac{E(R_n)}{n^{\frac{\beta}{\beta+1}}} \geq \frac{c^\beta}{\alpha} + \lim_{n \to +\infty} \frac{1}{n} \sum_{t=1}^{n} \frac{1}{c} \frac{\beta}{\beta+1} \left(\frac{n}{t}\right)^{\frac{1}{\beta+1}},$$

$$= \frac{c^\beta}{\alpha} + \frac{1}{c} \frac{\beta}{\beta+1} \int_0^1 \frac{du}{u^{\frac{1}{\beta+1}}},$$

$$= \frac{c^\beta}{\alpha} + \frac{1}{c} \geq \frac{\beta+1}{\beta} \left(\frac{\beta}{\alpha}\right)^{\frac{1}{\beta+1}},$$

the minimum being reached for $c = (\alpha/\beta)^{\frac{1}{\beta+1}}$.