[Reviews · NeurIPS 2013]

Submitted by Assigned_Reviewer_2

This paper consider the infinitely-armed bandit problem with Bernoulli distributions and a uniform prior over the parameters. The authors present a fundamentally new strategy and they show that on average it obtains a cumulative reward of order at least n - \sqrt{2 n} (in the known horizon case). They also show a matching lower bound and they extend the results to the anytime setting and more general priors.

I strongly vote to accept this paper. It is refreshing to see a truly new strategy to such a fundamental problem. I have a few minor comments:
- Could you try to derive finite-time results? At least it would be important to obtain the rate in m.
- In the lower bound could you remove the assumption that if you get a 1 then you stay on the same arm?
- The related work section could have been slightly expanded, perhaps a general reference to the bandit literature would help.
Summary: Truly new strategy for a fundamental problem.

Submitted by Assigned_Reviewer_4

The paper investigates a bandit model in which the rewards of the arms follow Bernoulli distributions and the number of arms is infinite. To tackle this problem, the authors proposed a 2-target algorithm in which an arm is chosen to be the best candidate if the length of its first run (i.e., the length of its first consecutive sequence of 1 rewards) has to be larger than a certain threshold l1(target 1), and the total length of its first m runs has to be larger than a certain value l2 (target 2), for a given m. By doing so, the authors could provide a \sqrt(2T) regret bound for the known horizon
case (T is the horizon), and 2\sqrt(T) for the case when the horizon is not known beforehand.
The former result rejects the conjecture of Berry et al,. which states that the lower bound of known horizon case is 2\sqrt(T).

I found the paper very easy to read. Indeed, I did not find any difficulty to follow the structure of the paper. I also found the proofs very interesting.

My only concern is that due to the restriction of the distributions, it might not attract much attention from the theoretical community. However, I believe that this model has a significant application domain, as Bernoulli distributions are quite frequent in many real world problems.
Summary: This paper investigates a very interesing model, advancing the state of the art by proposing a new algorithm that provides tight results on the regret bound.


Submitted by Assigned_Reviewer_5

The main contribution of the paper is to close a constant gap between the lower and upper bounds of the asymptotic regret in an infinite-armed bandit setting where the arms are drawn from a uniform distribution and the length of the horizon is known. The result is extended for unknown horizons and more generic distributions, at the expense of loosing the tightness between the lower and the upper bounds.

Quality
Algorithm 1 is an interesting addition to the literature. Its upper bound matches the lower bound proven in [2]. The algorithmic idea is novel and non-obvious and the regret proof is sound. It is important to note that the algorithmic strategy is only of theoretical interest at this point, as its regret guarantee holds when both n (the horizon length) and m (the number of "runs" of an arm) go to infinity.

My first question is whether you tried to get an upper bound for m in terms of n so that we can get a more informative result than Proposition 1? Such a result would help get rid of m as a parameter if one would actually want to use the algorithm in an applied setting.

One weakness of the paper is the lack of at least basic numerical results about the finite time behavior of Algorithm 1. Even though the result is asymptotic, it is interesting to understand how it behaves for a well defined horizon length (as discussed on page 2105 in [2] for example). While in section 5, the authors compare the empirical regret of the algorithm with the asymptotic lower bound, it would be interesting (since it might have an impact in an applied setting) to verify how the algorithm compares with the strategies introduced in [2] or [6] for example.

My second question is about the statement that the lower bound in [2] is incorrect (line 064). First, the authors mention Theorem 2 in [2], but Theorem 2 in the referred paper is about the upper bound for the k-failure strategy. So I am assuming they talk about the Theorem 3. The note at lines 102-105 mention an assumption from the lower bound theorem in [2] that I was unable to locate in that paper. As a consequence, I am not convinced that the original lower bound proof is incorrect. Could you please clarify? Also, in the case the paper is accepted and the lower bound is indeed incorrect, please add a more detailed description of the error in the appendix.

Clarity
The paper is well written and relatively easy to follow.

Regarding the organization, I would recommend decreasing the space allocated to the description of Algorithm 2 as it is pretty similar at a high level to Algorithm 1 and it clearly doesn't need 3/4 of a page to understand.

Originality and Significance
I am not fully convinced of the significance of the paper (at least from the point of view of being published in a conference like NIPS). While I think it is important that a gap is closed, I would like to understand the impact of the algorithm either from a more applied perspective or get some insights with respect to where it could be used from a theoretical perspective. That being said, to the best of my knowledge, the algorithmic ideas are novel and interesting and they might be of use for other bandit algorithms.
Summary: While I am not convinced about the importance of the results, the paper is an interesting contribution to the bandit literature as it describes a novel algorithmic strategy that is optimal in a simple infinite-armed bandit setting.
Author Feedback

Author rebuttal: We thank the reviewers for their useful comments. Here are our responses to their questions and concerns:
* Finite-time analysis: The proof of Proposition 1 is in fact based on a finite-time analysis. To get a simple, explicit upper bound on the regret as a function of n and m, it is indeed sufficient to replace the term (l2-j)! / (l2-l1-j)! by either l2! / (l2-l1)! or (l2-m+1)! / (l2-l1-m+1)! when the corresponding sum in j appears in the denominator or in the numerator, respectively, of the terms of the right-hand side of inequality (1). We plan to include the resulting expression in the statement of Proposition 1, as suggested by the reviewers.
* The "replay-upon-success" assumption: As pointed out by one of the reviewers, our proof of the lower bound is based on the assumption that any arm giving a success at time t is replayed at time t+1. Although this assumption is probably not critical, we have not yet been able to relax it.
* The incorrect argument in the lower bound of Berry et. al.: First, we indeed refer to Theorem 3 in [2] and not to Theorem 2; we thank the reviewer who pointed out this typo. Next, we should have referred to an independence "argument" instead of "assumption" in our text; this may have confused the reviewer. There is indeed no explicit assumption of independence. But the proof is based on the implicit assumption that the number of explored arms and the mean rewards of these arms are independent random variables. This is clearly incorrect, as shown by our simple example of the 1-failure policy in footnote 1 and, more explicitly, by the incorrect lower bound derived for a beta distribution in footnote 2.
* Other comments: The final version of the paper, if accepted, would devote more space to the numerical results, with a comparison with existing strategies like the run policy, and to the related work. We may indeed reduce the description of Algorithm 2 to get additional space.